# Highly thermostable carboxylic acid reductases generated by ancestral sequence reconstruction

Adam Thomas[1,2], Rhys Cutlan [1,2], William Finnigan[2], Mark van der Giezen [2,3] & Nicholas Harmer [1,2]*

Carboxylic acid reductases (CARs) are biocatalysts of industrial importance. Their properties, especially their poor stability, render them sub-optimal for use in a bioindustrial pipeline. Here, we employed ancestral sequence reconstruction (ASR) – a burgeoning engineering tool that can identify stabilizing but enzymatically neutral mutations throughout a protein. We used a three-algorithm approach to reconstruct functional ancestors of the Mycobacterial and Nocardial CAR1 orthologues. Ancestral CARs (AncCARs) were confirmed to be CAR enzymes with a preference for aromatic carboxylic acids. Ancestors also showed varied tolerances to solvents, pH and in vivo-like salt concentrations. Compared to well-studied extant CARs, AncCARs had a $T_m$ up to 35 °C higher, with half-lives up to nine times longer than the greatest previously observed. Using ancestral reconstruction we have expanded the existing CAR toolbox with three new thermostable CAR enzymes, providing access to the high temperature biosynthesis of aldehydes to drive new applications in biocatalysis.

[1] Living Systems Institute, Stocker Road, Exeter EX4 4QD, UK. [2] Department of Biosciences, Geoffrey Pope Building, Stocker Road, Exeter EX4 4QD, UK. [3] Present address: Centre for Organelle Research, University of Stavanger, Richard Johnsens gate 4, Stavanger 4021, Norway. *email: N.J.Harmer@exeter.ac.uk

Many industries are placing increasing emphasis on achieving carbon neutral manufacturing. For the chemical industry, the sustainable catalysis of high-value chemicals through enzyme cascades ("green chemistry") is a key opportunity[1,2]. Enzymes generally provide high yields with few side products and do so at mild reaction conditions. Enzymes therefore mitigate the production of excessive chemical waste and the use of toxic catalysts, while also reducing energy and solvent usage[3]. Nevertheless, enzymes are still poorly represented in the chemical synthesis market[4]. Enzymes are generally highly evolved towards their biological role in vivo, and rarely have properties optimized for a green chemistry application. Limited enzyme stability, restricted substrate ranges, substrate flux sinks, and low turnover rates are common barriers to success[5–9].

Carboxylic acid reductases (CARs; E.C. 1.2.1.30) are a family of enzymes with increasing relevance to green chemistry. They catalyze the reduction of an aliphatic or aromatic acid to the respective aldehyde, using ATP and NADPH as cofactors[10,11]. This reaction is otherwise challenging to achieve chemically or biochemically. Consequently, CARs are being used in biotechnology for the enantiopure biosynthesis of intermediates in enzyme cascades. Examples of these include biofuels[12,13], replacement petroleum-based intermediates[14], pharmaceutical building blocks[15], cosmetics[16], and flavorings (e.g. vanillin)[17].

There are currently four identified CAR subgroups: Subgroup I make up CARs of bacterial origin, while type II–IV make up CARs discovered in a broad spectrum of fungi[14,18]. CAR subgroup I can be further split into five families, of which family CAR1 (the focus of this study) is the best characterized. CARs consist of three distinct domains: an adenylation (A)/thiolation (T) domain, a phosphopantetheine (PPT)-binding domain, and a reductase (R) domain (Supplementary Fig. 1)[19]. The prevailing model for carboxylic acid reduction suggests the CAR reaction proceeds in four steps. The reaction is initiated in the A/T-domain by a nucleophilic attack of the acid on ATP to form an AMP-acyl ester intermediate. Structural determination of CAR fragments indicate that an A/T-subdomain undertakes a 165° rotation characteristic to the superfamily to which this domain belongs (CL0378)[16,19]. Additionally, the PPT-binding domain undertakes a 75° rotation relative to the A/T-domain[19]. This dynamic re-orientation of the subunits relative to one another presents the AMP-acyl intermediate to the PPT, which displaces AMP to form a PPT-acyl thioester intermediate. This intermediate is then passed to the reductase domain. Here, the intermediate is reduced by NADPH to release an aldehyde product (current model described in Supplementary Fig. 1).

CAR1 family CARs have demonstrated diverse substrate ranges, with activity against over 100 carboxylic acids[11,18], including both aromatic acids[10,20,21] and aliphatic acids[10]. This diverse substrate range and apparent substrate plasticity highlights CARs' broad potential in green chemistry. However, CARs lack some desirable properties. It has been highlighted that isolation of CARs with improved thermostability is an important goal to improve the CAR toolbox[11]. Green chemistry pipelines benefit from operating at increased temperatures to improve substrate solubility and reaction rates while mitigating risks of contamination and costs from cooling[22–24]. Additionally, stable enzymes can often be operated longer than their unstable counterparts, improving per-enzyme productivity per batch reaction, lowering the cost of the enzyme relative to the product[3,22]. Other desirable biocatalytic properties include solvent tolerance, broad substrate ranges, and ready evolvability. We previously reported that well characterized extant CARs (ExCARs) are barely suitable for reactions above 37 °C. The most stable ExCAR (from *Mycobacterium avium*) loses activity rapidly above 49 °C, and retains 50% of activity after incubation for 30 min at 48 °C[25]. ExCARs

also show short half-lives at 37 °C and will likely present a huge metabolic burden for biofactory strains[10]. Therefore, the current state of the CAR toolbox only services batch biocatalysis, which significantly reduces their scale-up potential, hampering their use in biotechnology.

Ancestral sequence reconstruction (ASR) is a popular tool to study the evolutionary histories of protein families. ASR studies of diverse protein families have identified emergent properties of ancestral proteins, including increased thermal stability and altered substrate specificities[26–29]. Consequently, a series of studies have used evolutionary histories to isolate sites of interest to engineer enzymes with novel functionality[30–34]. When used as an engineering tool, ASR has produced enzymes with improved stability[35], substrate ranges[36], or both[37]. ASR differs from other engineering methods as it generates new sequences based upon probabilistic searches of non-conserved functional space, giving each output a high likelihood of being functional given an accurate sequence alignment input. Given enough variation in the input dataset, resulting ancestors can often vary considerably from extant sequences (<30%). This allows for the discovery of beneficial mutations not accessible by other methods, including coordinated sets of mutations. These can modify traits determined by protein-wide sequence states, including stability under thermal or other stresses.

Notably, all studies to date focusing on ASR for engineering explore ancestral sequence space use a single reconstruction algorithm. Additionally, most available algorithms output "posterior probabilities" at each residue, providing a sequence space representing putative ancestors around a point in sequence space[38,39]. Variation within this space is a resource of both sequence and functional diversity[40]. When sampling through these posterior probabilities, there is no "ruleset" dictating the best probability cut-off to efficiently explore space—an issue that has presented in other alignment-based engineering methods (e.g. consensus alignment[41]). To avoid this issue, we instead explored the algorithmic variation within the ASR toolbox as a source of sequence and functional diversity by deriving the most likely sequence from multiple maximum likelihood-based reconstruction algorithms. Each algorithm differs subtly, and therefore will output a different, absolute sampling of ancestral sequence space when given the same problem[38,39,42,43].

Here, we demonstrate the use of ASR to identify three ancient actinomycete CAR biocatalysts that display a 16–35 °C shift in thermal stability compared to ExCARs. The three alternative putative ancestral proteins showed similar substrate ranges and refined substrate preferences to ExCARs. Comparison of the output from different reconstruction algorithms showed dramatic variations in tolerance to loop-based conditions between the putative ancestral proteins, including tolerance to in vivo-like salt concentrations, pH, and protic and aprotic solvents. This study represents one of the largest proteins to have been reconstructed successfully by ASR to date, and the first reconstruction of an enzyme with four mechanistic steps to our knowledge. This further demonstrates ASR's potential application to biotechnology and green chemistry.

## Results

**Ancestral reconstruction of CARs produces functional enzymes**. We previously reported a dataset of 124 CAR homologs identified from the CAR1 family[10]. Of this dataset, 48 sequences representing distinguished clades containing a single genus were used to produce a phylogeny broadly covering CAR sequence space. An alignment of the 48 CAR sequences was created in MUSCLE (Supplementary Fig. 2). Removal of highly divergent regions in the alignment was conducted with the Gblocks

algorithm[44]. ProtTest estimated the best fitting model of amino acid substitution for this alignment to be WAG + I + G[45,46]. As only CAR1 enzymes had been reported at the point of reconstruction, we aimed to reconstruct the ancestors of their best represented families, from Mycobacteria, Nocardia, and Streptomyces. To construct the phylogeny, we therefore treated well established sequences from *Tsukamurella* and *Segnilliparus* (here the *Tsukamurella* clade) as paralogues, providing an outgroup to the Mycobacteria, Nocardia, and Streptomyces clades. The

resulting phylogeny was well supported throughout (Fig. 1a; Supplementary Fig. 3).

Within recent literature, marginal ancestral protein reconstruction has been shown to introduce novel functional properties into proteins[37,46,47]. As ancestral proteins typically trend towards increased stability when sampling from more ancient nodes[40], we reconstructed the most recent common ancestor of the Nocardia, Streptomyces, and Mycobacterial CARs. To explore differences reconstruction algorithm choice had on the sequence and

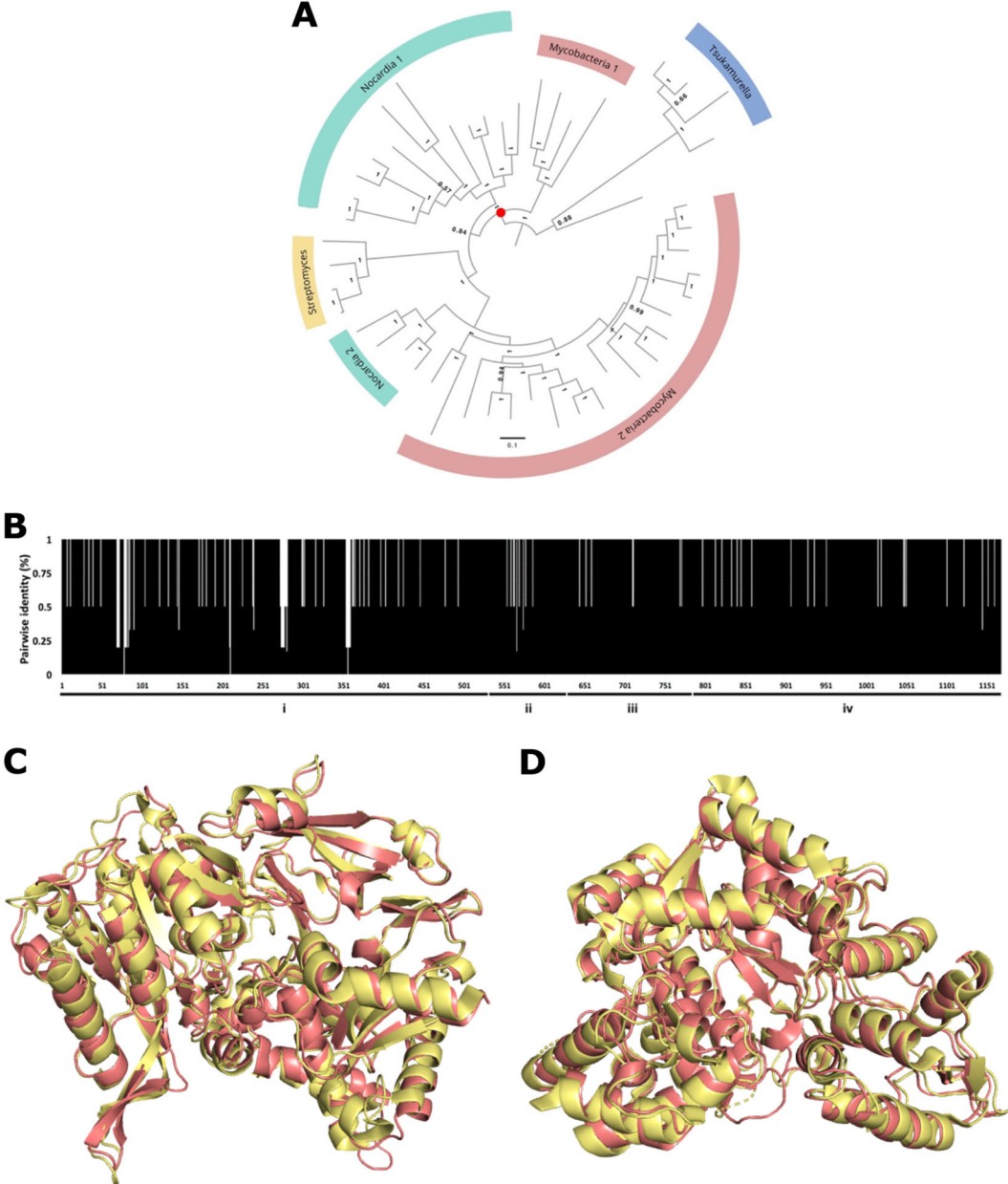

**Fig. 1** Bayesian inference of actinomycete CAR phylogeny and ASR. **a** The phylogeny of the CAR1 group was constructed with the *Tsukamurella* clade constrained to be the outgroup. The tree was configured in FigTree V1.4.3 and edited with Gravit designer. The scale bar represents amino acid changes per site. Node weights represent the posterior probability of a given node calculated from the MCMCMC analysis, with 1 being unequivocal. Red circle represents the target node for ancestral reconstruction. **b** Identity barcode displaying the pairwise identity over 1168 amino acid sites between the four ancestors. *X*-axis denotes residues 1–1168 sequentially, *y*-axis denotes pairwise identity at a site. Black bars denote pairwise identity (%) at each site. AncCAR-F and PF are 1161 aa in length, and AncCAR-A and PA are 1153 aa in length. Alignment data between ancestors were obtained in Geneious using MUSCLE and modified in Microsoft Excel. Domains are highlighted: (i) Adenylation domain; (ii) phosphopantetheine-binding di-domain 1; (iii) phosphopantetheine-binding di-domain 2; (iv) reductase domain. **c** Model of the AncCAR-PF adenylation domain superimposed on extant CAR structure 5MST. Structures: yellow: 5MST; red: AncCAR-PF. **d** Model of the AncCAR-PF reductase domain superimposed on ExCAR structure 5MSO. Structures: yellow: 5MSO; red: AncCAR-PF. Equivalent images for the other AncCARs are given in Supplementary Fig. 6. Images produced using PyMOL.

property space sampled in the ancestor, three marginal reconstruction algorithms with optimized likelihood scores were used: FastML[39], PAML[38] and Ancescon[43]. This produced four putative ancestral proteins: AncCAR-A (Ancescon); AncCAR-F (FastML); and PAML variants with gaps reconstructed by cross-mapping from the other two algorithms producing AncCAR-PA and AncCAR-PF, respectively (Supplementary Fig. 4). AncCARs possessed 95.1% pairwise identity, and 91% conservation across the four proteins, with much of the variation being held in the adenylation domain (Fig. 1b). Their identity to ExCARs ranges between 55% and 76%. To explore whether algorithmic variation was merely sampling variation from posterior probabilities, an ancestor was derived from the PAML output where the most probable residues were substituted with the second most probable residues where the latter showed a probability over 30% (AncCAR-P30). The resulting protein shared 93.6% pairwise identity to the algorithm-derived AncCARs. We observed that algorithm-derived variation differed considerably from the posterior probability-derived variation (Supplementary Fig. 5a). Compared to the most likely AncCAR-P sequence, only 16% and 22% of total derived variation was shared between AncCAR-P30 and AncCAR-A, and AncCAR-P30 and AncCAR-F respectively (Supplementary Fig. 5b). To give confidence that reconstructed ancestral CARs were accurate representations of CAR enzymes, we modeled AncCARs using homology modeling to crystal structures 5MST, 5MSD, 5MSP, and 5MSO. Comparing all ancestor models to all extant structures, the average root mean squared deviation (rmsd) of alpha-carbon atom position is 0.86 ± 0.32 Å between ancestral models and extant adenylation domains, and 1.01 ± 0.15 Å between ancestral models and extant reductase domains, suggesting a good fit for each model (Fig. 1c, d; Supplementary Fig. 6; Supplementary Table 1). Comparison of the models to experimental crystal structures shows that most of the variation between ancestors occurs in surface loop regions (Supplementary Fig. 7). Each AncCAR could be expressed in, and readily purified from *E. coli* to between 3 and 7 mg enzyme per liter (Supplementary Fig. 8), similarly to what we obtain from ExCARs. The ancestral CAR proteins demonstrated some protease sensitivity. However, in comparison to ExCARs, they were more resistant to limited proteolysis by common proteases (Supplementary Fig. 9).

**Substrate range of AncCARs.** Assays of AncCAR activity were performed in HEPES instead of the canonical CAR buffer system Tris, as HEPES is more suited to pH 7.5, and Tris was found to inhibit AncCAR activity above 50 mM (Supplementary Fig. 10; similar results were observed for ExCARs). AncCARs were screened for activity on 21 aromatic and aliphatic fatty carboxylic acids at 5 mM concentrations. No significant activity could be detected for AncCAR-F on any of these substrates across several protein preparations. This protein was therefore eliminated from further kinetic analyses. The other three AncCARs show equivalent substrate ranges to one another across all substrates tested. Ten of the 21 substrates, including 9 aromatic carboxylic acids and 1 aliphatic carboxylic acid, showed a statistically significant NADPH turnover ($P \leq 0.001$) compared to background rate for at least 2 of the 3 ancestors (Supplementary Fig. 11). A subset of these are shown in Fig. 2a.

Kinetic analysis of AncCAR activity was first conducted on NADPH and ATP in the presence of 5 mM (*E*)-3-phenylprop-2-enoic acid (Supplementary Fig. 12). For NADPH, AncCAR $K_M$ values were similar to those derived from ExCARs. On the other hand, observed $K_M$ values for ATP were between 10 and 100 times lower than values derived for ExCARs (ref. [10]; Table 1).

This suggests ATP binding is considerably tighter in the AncCARs.

AncCAR kinetics on substrates showing significant activity from background were then tested in saturating NADPH and ATP levels (Supplementary Fig. 13a). The Michaelis constant of all AncCARs was typically determined to be approximately 10-fold higher than ExCARs[10] (Supplementary Fig. 13b). All AncCARs showed strong activity on canonical substrates: benzoic acid and its derivative 4-methylbenzoic acid. AncCARs have a clear preference for substrates with electron-rich conjugated carboxyl groups, with turnovers being among the highest across all tested substrates for all ExCARs[11] (Table 2). For example, AncCAR-PA turnover of 3-phenylpropionic acid is the highest turnover rate observed for any substrate across all four CAR subgroups, 1.5-fold higher than that of any substrate reported for the CAR1s (468 min⁻¹; Table 2). Finally, while AncCARs are active on octanoic acid, AncCAR preference for fatty acid substrates is attenuated compared to ExCARs, with no activity seen for canonical 3-C and 5-C aliphatics. Octanoic acid was turned over by AncCARs at rates comparable to ExCARs; however, each enzyme showed an approximately 100-fold higher $K_M$ (Table 2). Octanoic acid also displayed substrate inhibition on AncCARs at high concentrations (Supplementary Fig. 13). The increased substrate $K_M$ reflects that CAR1 family likely evolved from fatty acyl CoA-ligases[10]. The CAR ancestor would likely be poorly adapted for most acid substrates.

In AncCAR homology models, we observed that the active site of the ancestors' adenylation domains appear to be slightly disordered compared to the extant structures[18] (Fig. 2b). This is most evident when comparing differences in a variable loop that stretches into the active site between positions 286 and 302. CAR models implicate these residues in hydrophobic interactions with the substrate, likely positioning it in catalytically favorable conformations during the adenylation step. The catalytically essential His315 is positioned as a rotamer away from the substrate, suggesting this residue has a large sampling space within the active site of the AncCARs. In the model of AncCAR-PF (Fig. 2c), this loop region is significantly shortened and is unable to contact the substrate. Comparison of inactive AncCAR-F to ancestor models and ExCAR structures showed no obvious structural or functional residue changes that explain the loss of activity (Supplementary Fig. 14).

**Ancestral CARs show dramatic increases in stability.** Many ancestral proteins have displayed increased resistance to temperature[35,40,46,48–52]. All AncCARs retained 50% activity following incubation at temperatures of >65 °C, compared to less than 50 °C for a stable extant protein. AncCAR-A is the most thermostable ancestor, and the most stable CAR protein reported to date, retaining 50% activity at around 70 °C (Fig. 3a; 50% activity retained for AncCAR-PA and AncCAR-PF at 65.1 and 65.4 °C respectively). Monitoring of AncCAR unfolding in real time with differential scanning fluorimetry[53,54] also corroborates that AncCARs are highly stable. All AncCARs showed the greatest rate of unfolding ($T_m$) between 67 and 68 °C (Fig. 3c). AncCAR half-life at 37 °C in 50 mM HEPES was monitored by assessing their activity on 5 mM (*E*)-3-phenylprop-2-enoic acid at intervals over a period of 10 days. AncCAR-A showed a short half-life of less than 41 h. This was of stark contrast to AncCAR-PA and AncCAR-PF, whose half-lives at 37 °C were between 168 and 216 h. AncCAR-PA and AncCAR-PF display the longest half-lives reported to date in CARs (Fig. 3d). These half-lives significantly exceed those of the CAR from *Mycobacterium phlei*, which was previously reported to have the longest half-life of ExCARs[10].

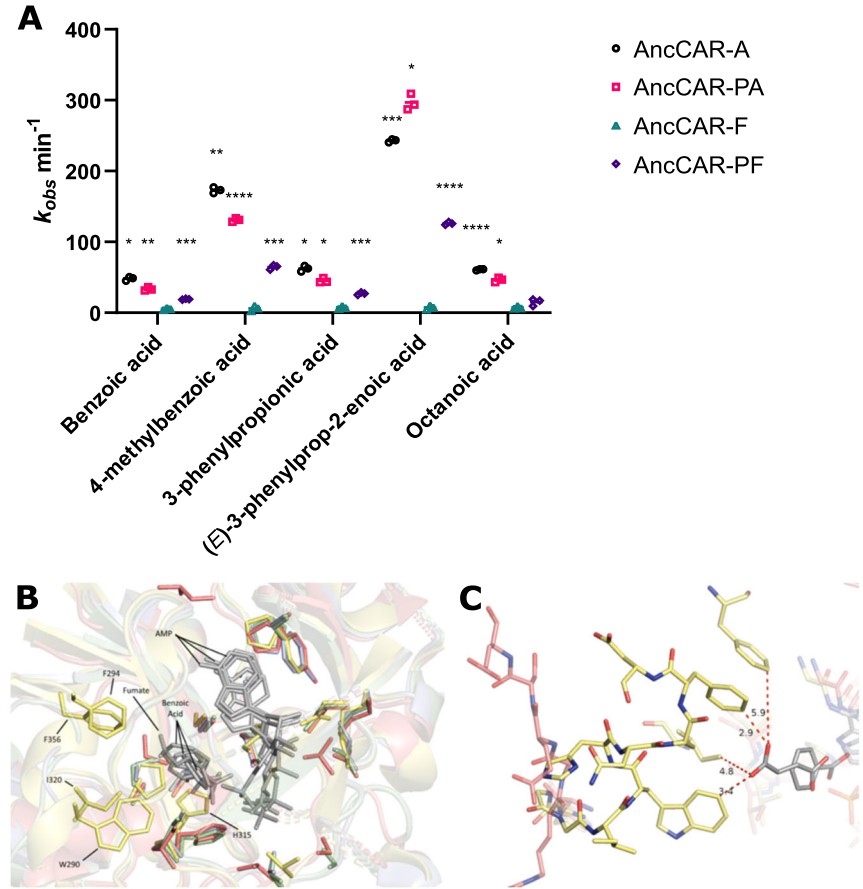

**Fig. 2 AncCARs have CAR-like substrate kinetics. a** Turnover of NADPH by AncCARs was measured with 24 unique carboxylic acids, of which 5 are shown here. Bar chart shows activity on canonical acid substrates at 5 mM. Each substrate was tested using three experimental replicates. Asterisks represent degrees of significance from *t*-test of each experiment compared to controls (* = $0.0001 < P \leq 0.001$; **$0.00001 < P \leq 0.0001$; *** = $0.000001 < P < 0.00001$; **** = $P \leq 0.000001$). Complete substrate screens are presented in Supplementary Fig. 13. **b** The active site structure of the adenylation domain. AncCARs A (green), PA (blue), and PF (red) are overlaid onto *S. rugosus* CAR (PDB ID: 5MST; yellow). Substrates are shown in gray. The residues lining the active site pocket of ExCARs (positions 246–250) are poorly resolved. **c** In AncCAR-PF, the highly variable loop between positions 286 and 302 of the adenylation domain of AncCAR-PF (red) does not interact with the substrate, in contrast to SrCAR (5MST; yellow). Images produced in PyMOL v.2.2.

**Table 1 Kinetic parameters for AncCARs and their cofactors.**

| | ATP | NADPH |
|---|---|---|
| **AncCAR-A** | | |
| $k_{cat}$ (min$^{-1}$) | 340 ± 2.7 | 386.4 ± 11.1 |
| $K_M$ (µM) | 76.8 ± 4.3 | 54.8 ± 5.1 |
| $k_{cat}/K_M$ (min$^{-1}$ µM$^{-1}$) | 4.4 ± 0.3 | 7.1 ± 0.7 |
| **AncCAR-PA** | | |
| $k_{cat}$ (min$^{-1}$) | 392.3 ± 11.2 | 482.2 ± 15 |
| $K_M$ (µM) | 69.1 ± 6.7 | 58.5 ± 5.0 |
| $k_{cat}/K_M$ (min$^{-1}$ µM$^{-1}$) | 5.7 ± 0.6 | 8.2 ± 0.8 |
| **AncCAR-PF** | | |
| $k_{cat}$ (min$^{-1}$) | 219.2 ± 3.1 | 230.6 ± 2.5 |
| $K_M$ (µM) | 42.9 ± 2.3 | 29.0 ± 1.2 |
| $k_{cat}/K_M$ (min$^{-1}$ µM$^{-1}$) | 5.1 ± 0.3 | 8.0 ± 0.3 |
| **ExCARs[10]** | | |
| $K_M$ (µM) | 64–84 | 24–36 |

Rates of AncCAR activity with ATP and NADPH were determined using a 12 point, 1.7× dilution series of substrate, with concentrations starting at 800 µM. Each concentration was investigated using three experimental replicates for each concentration point. Data were fitted to the Michaelis–Menten equation using GraphPad v.7.0. Graphs are shown in Supplementary Fig. 12. Errors show SEM

Importantly, biocatalysts are used for both in vitro and in vivo bioindustrial pipelines. Robust biocatalysts are therefore required to function in the highly ionic environments demanded by in vivo bioconversions. Ionic solutions can have either stabilizing or destabilizing effects on enzymes[55]. To better characterize AncCARs for use in the CAR toolbox, their thermostability was assessed in a buffer simulating the ionic environment inside a *Saccharomyces cerevisiae* cell[56]. In these potentially challenging conditions, AncCAR-PA was the least thermostable ancestor, with an A$_{50}$ of 45 °C—a 20 °C decrease over incubation in standard in vitro assay conditions. AncCAR-A showed a 16 °C decrease in stability over the salt-free condition, presenting an A$_{50}$ of approximately 54 °C, losing activity in a near linear fashion from around 40 °C. AncCAR-PF is the only ancestral protein observed to be halotolerant at temperature, with an equivalent A$_{50}$ to the salt-free condition at 65 °C (Fig. 3b). To confirm it was the presence of salt that was effecting stability in AncCARs, we repeated the experiment in the presence 500 mM NaCl. Equivalent destabilizing effects to the in vivo-like conditions were observed (Supplementary Fig. 15).

**AncCARs vary in their loop-based properties**. Salt-tolerance has been proposed as a "loop-associated" trait, where net surface

**Table 2 Kinetic parameters of AncCARs with key substrates.**

| | Benzoic acid | 4-Methylbenzoic acid | 3-Phenylpropionic acid | (*E*)-3-phenylprop-2-enoic acid | Octanoic acid |
|---|---|---|---|---|---|
| **A** | | | | | |
| $k_{cat}$ (min$^{-1}$) | 149.1 ± 7.1 | 398.4 ± 13.9 | 327.3 ± 16.9 | 203.5 ± 3.8 | 302.6 ± 19.2 |
| $K_M$ (mM) | 61.2 ± 5.6 | 6.5 ± 0.4 | 33.0 ± 3.4 | 0.9 ± 0.06 | 8.7 ± 1.1 |
| $k_{cat}/K_M$ (min$^{-1}$ mM$^{-1}$) | 2.4 ± 0.3 | 61.3 ± 4.3 | 9.9 ± 1.1 | 226.1 ± 15.7 | 34.9 ± 4.9 |
| **PA** | | | | | |
| $k_{cat}$ (min$^{-1}$) | 176.4 ± 7.7 | 146.6 ± 6.8 | 468.1 ± 36.7 | 396.4 ± 5.6 | 344.1 ± 27.1 |
| $K_M$ (mM) | 81.4 ± 6.2 | 5.5 ± 0.5 | 68.2 ± 8.5 | 4.0 ± 0.1 | 11.0 ± 1.6 |
| $k_{cat}/K_M$ (min$^{-1}$ mM$^{-1}$) | 2.2 ± 0.2 | 26.7 ± 2.7 | 6.9 ± 1.0 | 99.1 ± 2.8 | 31.3 ± 5.1 |
| **PF** | | | | | |
| $k_{cat}$ (min$^{-1}$) | 71.8 ± 2.7 | 61.9 ± 2.8 | 325.6 ± 26.3 | 193.8 ± 8.0 | 169.0 ± 13.7 |
| $K_M$ (mM) | 79.9 ± 5.4 | 7.0 ± 0.5 | 98.7 ± 11.5 | 3.8 ± 0.3 | 11.1 ± 1.7 |
| $k_{cat}/K_M$ (min$^{-1}$ mM$^{-1}$) | 0.9 ± 0.1 | 8.8 ± 0.7 | 3.3 ± 0.5 | 51.0 ± 4.5 | 15.2 ± 2.6 |

(C) Rates of AncCAR activity with various aromatic and aliphatic compounds were determined using an 8 point, 1.7× dilution series of acid from near saturation in 125 mM HEPES. Each concentration was investigated using three experimental replicates. Data were fitted to the Michaelis–Menten equation using GraphPad v.7.0. Graphs are shown in Supplementary Fig. 12. Errors show SEM

charge effects solvent penetrance[55]. Following our observation that much of the variation between ancestors is loop-based, we further investigated AncCAR's resistance to other "loop-associated" conditions. Solvent tolerance is a common industrially relevant loop-associated property desired in biocatalysts. We assessed the AncCARs' solvent tolerance in a range of protic and aprotic solvents at increasing solvent concentrations in comparison to example ExCARs (Supplementary Table 2). There is no consistent trend observable between all ancestors on all solvents. AncCAR-A is the least solvent tolerant enzyme for all solvents besides DMSO and methanol. For all solvents besides acetone, AncCAR-PF is the most solvent tolerant, retaining 50% activity in the presence of over 25% methanol. Ancestors show the greatest variance to tolerance in methanol, with AncCAR-PF showing considerable increases in tolerable concentration of solvent compared to AncCAR-A (89%) and AncCAR-PA (119% increase). In protic solvents, AncCAR-PF performed similarly to the most solvent tolerant ExCAR. In contrast, in aprotic solvents, the AncCARs generally showed greater activity than ExCARs. This was particularly so in DMSO (standard deviation 0.3%), with the ancestral proteins retaining 86–92% activity in DMSO (v/v) solvent (Fig. 4a; Supplementary Table 2), compared to 67–74% for the ExCARs. A wide pH tolerance for industrially relevant enzymes is another highly desirable loop-associated trait. All AncCARs displayed no loss of activity between 6.0 and 9.0 pH units (Fig. 4b). AncCAR-A lost activity in alkaline conditions above pH 9.0, whereas AncCAR-PF and AncCAR-PA maintained 100% activity up to pH 10.0. All ancestral CARs show a decrease in activity below pH 6.0 (p$K_1$ ≈ 5 for all three enzymes). However, this feature is shared with ExCARs, with the CAR from *Mycobacterium phlei* (MpCAR) showing even greater pH tolerance than AncCAR-PF, the most pH tolerant of the AncCARs (50% activity between pH 5.01 and pH 11.56 for AncCAR-PF, compared to pH 4.3 to 11.8 for MpCAR).

## Discussion

Protein engineering for the optimization of application specific properties in enzymes is integral to the future green chemistry market. Limited understanding about the sequence–function relationship in biocatalysts presents a significant challenge for synthetic biology. This is exemplified by the CARs. Single amino acids that regulate CAR function and selectivity are starting to be uncovered, including active site point mutants that modulate substrate turnover[18]. Nevertheless, at present without significant innovation in the protein engineering field the semi-rational

engineering of CARs with high-throughput approaches would remain prohibitively expensive. Furthermore, CARs with improved stability have been highlighted as an important potential addition to the CAR toolbox[11]. However, there are no defined rules available to guide the rational engineering of thermostability in any enzyme, let alone one as complex and poorly understood as CARs[52].

Here, we aimed to sample ancient sequence space using multiple ASR algorithms to engineer stability into CARs. CARs present as challenging targets for ASR: they are large (>1100 amino acids) and dynamically complex proteins. While large or multidomain proteins have been successfully reconstructed (e.g. estrogen receptors —~600 aa[57]; an eight domain titin fragment—~700 aa[58]; and the six domain factor VIII—~2300 aa[59]), CARs undertake four catalytic steps, including two large-scale domain reorientations[10,19]. CARs consequently represent a particularly interesting subject for ancestral reconstruction. In the first instance, it is therefore surprising that all four reconstructed enzymes could be readily expressed and purified in *E. coli* (Supplementary Fig. 8). It is even more surprising that three of the four putative ancestors were functional CAR-like enzymes, showing unambiguous CAR activity against a range of standard CAR substrates (Fig. 2; Supplementary Figs. 11 and 13). AncCARs identified were highly conserved (91% identity; Fig. 1b). Despite such high conservation, a broad functional space was identified. In particular, despite over 95% sequence identity between the AncCAR-F and AncCAR-PF proteins, the AncCAR-F was non-functional, while the AncCAR-PF showed broad activity. Although the PAML and FastML algorithms are highly similar, they have subtle differences that are expected to result in some sequence differences in ancestors given the length and sequence diversity of the CARs. Many of the non-identical residues were at sites that the algorithms considered equivocal. Our use of multiple reconstruction algorithms also allowed for the sampling of sequence space in an empirical manner, unique from the obtuse posterior probability-based sampling methods (Supplementary Fig. 5). Homology modeling suggests that variation between the ancestors is concentrated at surface loops, mostly within the A domain (Fig. 1b; Supplementary Fig. 7b). Loops are flexible regions within a protein that can exhibit large degrees of motion, are often tolerant to amino acid substitution and are a key determinants of protein stability[60,61]. We observe that AncCARs vary in their loop-dependent properties, with variation in their tolerance to in vivo-like salt concentrations (Fig. 3b), in their activity in protic and aprotic solvents (Fig. 4a; Supplementary Table 2), and in their tolerance to alkaline conditions

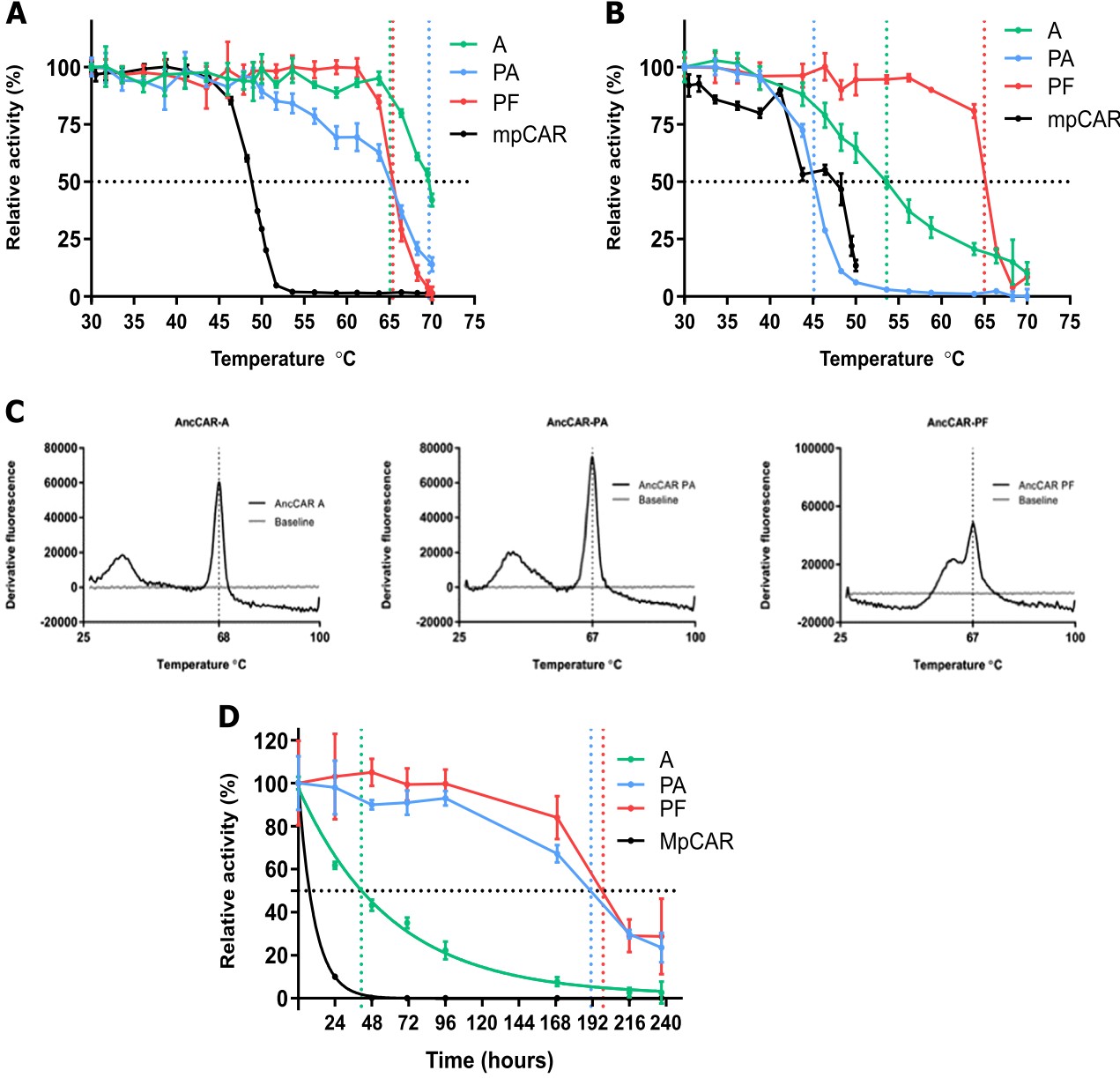

**Fig. 3** AncCARs are thermostable enzymes. **a** AncCARs and the CAR from *Mycobacterium phlei* (MpCAR) were incubated in 50 mM HEPES at temperatures from 30 to 70 °C for 30 min. Each point represents the rate of NADPH oxidation in 5 mM (*E*)-3-phenylprop-2-enoic acid at temperature relative to the rate of NADPH oxidation in 5 mM (*E*)-3-phenylprop-2-enoic acid at 30 °C. Black dotted horizontal line represents 50% activity. Colored vertical dotted lines represent temperature at which $A_{50}$ is reached. **b** AncCARs have environment-dependent temperature resistance. AncCARs were incubated in in vivo-like ionic concentrations that model the internal environment of a *S. cerevisiae*[56] cell at temperatures from 30 to 70 °C for 30 min. Data were determined as in panel **a**. Black dotted horizontal line represents 50% activity. Colored vertical dotted lines represent temperature at which $A_{50}$ is reached. **c** Differential scanning fluorimetry. AncCARs were incubated in HEPES and analyzed from 25 to 100 °C. Thermal shift curves were drawn from raw DSF data in GraphPad. **d** To assess half-life at 37 °C, AncCARs and MpCAR were incubated at this temperature over a period of 10 days. Relative activity versus a zero-time point was assessed by activity on 5 mM (*E*)-3-phenylprop-2-enoic acid. Black dotted horizontal line represents 50% activity. Colored vertical dotted lines represent time taken to reach 50% enzyme activity. Error bars represent standard error, in all cases calculated from three experimental replicates.

(Fig. 4b). Such conditions modify the sum of zwitterionic states across the protein surface, causing repulsive forces within the protein's loop regions. In turn, increased repulsion of loops expose the hydrophobic core of the protein to bulk solvent[55,62,63]. As loop-based regions are resistant to the deleterious effects of mutation, they are more likely to vary in the extant dataset, allowing ASR-based searches of ancient sequence space to capture

this variation at the functional level. These results highlight the potential of ASR as an engineering tool even for large, complex biomolecules that are otherwise less tractable for protein engineering.

The different ancestral reconstruction algorithms that we used apply subtly different gapping regimens. These are likely to partly explain the variation in both loop-based properties and reaction

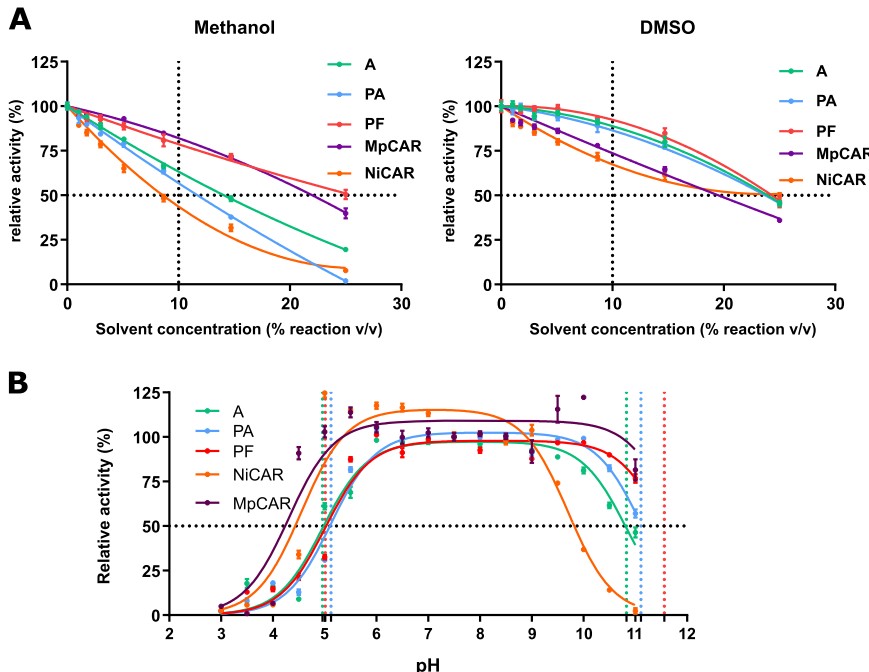

**Fig. 4** AncCAR tolerance to loop-dependent environmental factors. **a** AncCAR and ExCAR activity on 5 mM (*E*)-3-phenylprop-2-enoic acid was assessed in aprotic and protic solvents by solvent titration from 25% (v/v). For all proteins on all solvents besides DMSO (A, 88.9%; PA, 86.5%; PF 92.4%) considerable activity is lost at 10% solvent concentration for AncCARs, while ExCARs also lose activity in DMSO (MpCAR, 74%; NiCAR, 67%). As DMSO had the smallest inhibitory effect and lowest variance between enzymes, 10% DMSO was chosen for further kinetic analyses. Graphs represent relative activity of each ancestral enzyme at increasing concentrations of solvent compared to 0% solvent. Data for all solvents can be found in Supplementary Table 2. **b** To assess the resistance of AncCARs folding to pH, AncCARs and ExCARs were incubated for 30 min in 0.5 pH increments between pH 3 and 11, before being assayed for their turnover of NADPH in the presence of 5 mM (*E*)-3-phenylprop-2-enoic acid relative to turnover at pH 7.5 (100%). Data were analyzed in GraphPad Prism 7.0. $pK_1$ and $pK_2$ values were calculated respectively as: AncCAR-A—4.96 ± 0.06 and 10.83 ± 0.06; AncCAR-PA—5.12 ± 0.05 and 11.11 ± 0.07; AncCAR-PF—5.011 ± 0.06 and 11.56 ± 0.11. NiCAR—4.55 ± 0.09 and 9.70 ± 0.09; MpCAR—4.3 ± 0.1 and 11.8 ± 0.3. Error bars represent standard error, calculated from three experimental replicates.

rates reported between ancestors. This is particularly so for the two PAML gapping variants (AncCAR-PA and AncCAR-PF). The majority of gaps occur in variable loops. These include a critical loop stretching into the adenylation substrate binding pocket. Altering loop lengths can modify structural flexibility, with a concomitant impact on stability as discussed above[61–63]. This highlights the importance of gap reconstruction in ASR studies. We would encourage other ASR users to attempt reconstruction with multiple ASR algorithms when working with alignments that contain gaps, to confirm that gap placement is coordinated between methodologies. In cases where the sequence identity is sufficiently high to eliminate any ambiguity in gap locations, the use of multiple algorithms may be less important. We would also encourage future ASR engineering studies to include consideration of gap placement to expand understand of the impact this has on obtainable property space.

This study expands on previous work investigating ASR's use as a protein engineering tool, confirming its tractability to the engineering of large, mechanistically complex multidomain proteins. Importantly, all functional CAR ancestors were found to be highly thermostable ($A_{50} > 65\,°C$) in simple buffer conditions (Fig. 3a). AncCAR-A, with around 50% activity retained after incubation at 70 °C, shows 15–35 °C greater thermostability than ExCARs[10,25]. However, ancestors showed considerable variation in their half-lives, with AncCAR-A losing 50% activity on just over 40 h, whereas AncCAR-PA and PF maintained at least 50% activity for over a week (Fig. 3d). As we are not aware of a highly thermostable CAR variant that exists in the CAR toolbox, ancient CAR enzymes provide much needed functionality, providing a

means to convert carboxylic acids into aldehydes within a high temperature biocatalysis. Overall, AncCAR-PF presents as an attractive, all-purpose CAR enzyme due to its extraordinarily hardy nature, and broad scale resistance to many challenging conditions. It is stable up to around 65 °C in both in vivo and in vitro conditions, it has a half-life of over a week, it has a pH range of 6.5 pH units and it is exhibits the highest tolerance to solvent in all tested cases besides acetone. These collective properties are highly desired in CAR enzymes due to the poor solubility of their aldehyde products, ensuring efficient coupling to downstream bioconversions. On the other hand, AncCAR-A and AncCAR-PA appear to be excellent biocatalysts for the production of cinnamic aldehyde derivatives. AncCAR-PA's turnover of 3-phenylpropionic acid is the highest turnover rate observed to date for any CAR from any family on any substrate.

Importantly, such enzyme improvements are of broad industrial relevance as they were achieved with free software, without the prerequisite of an experimental structure, and without having to produce or screen a library of variants. ASR's delivery of large stability increases will therefore offer a cost and time-saving opportunity in current protein engineering pipelines. We further anticipate that ASR will not replace existing engineering pipelines, but instead act as a front-end process. Enzymes with increased stability "smooth" the sequence-function landscape. This occurs as stable enzymes can permit the introduction of destabilizing mutations without cost to enzyme fitness, thus improving mutational robustness and introducing new avenues for property discovery[23,64,65]. It therefore follows that ancestral enzymes could be more easily engineered for improved or refined

activities[66]. Being able to rapidly "strip back" enzymes to a more plastic molecule may provide improved avenues for more complex protein engineering pipelines.

In terms of experimental design in ASR, our observation of a rich ancestral property space informs important considerations. It is commonplace in today's ancestral reconstruction literature, whether focused on engineering or on evolution, that ancestors are constructed from nodes in a single lineage or small number of lineages to understand their properties[50,67,68]. To our knowledge, only benchmarking studies have assessed the difference between algorithms, and have done so on a very small number of enzyme targets[42,69]. However, our work shows that the properties of sequences derived from different algorithms differ based on ancestral reconstruction method, yet no algorithm can be argued to provide more confident representation of ancestral space. Therefore, in future ASR work, comparisons between ancestors made with different algorithms might provide better insight into ancestral property space. Additionally, we show that ancestors exhibit vastly different stability profiles, dependent on whether the proteins are being assayed within an in vitro and in vivo-like environment. If one was to conclude that thermostable proteins confer a high stability of ancient life, one must prove that this is the case in in vivo conditions, as the limits of protein stability within the cell environment define the environmental limits in which an organism can survive[70]. To our knowledge, all ASR studies that address the temperature environment of early life only test their proteins using in vitro conditions[26,29,40,48,49,51]. We therefore encourage caution be taken when drawing conclusions about a protein's environment based on in vitro stability alone as well as conclusions drawn from one representation of ancestral space at a given node.

## Conclusion

ASR offered a very attractive solution for engineering CARs, as their complexity makes them intractable to conventional protein engineering. This methodology greatly increases the likelihood of obtaining functional sequences, as every extant sequence referenced already contains permitted residues at each position. Here, using ASR, we have successfully engineered three functional CAR enzymes with novel properties tractable to biotechnology. All three ancestors bring valuable properties to the CAR toolbox, providing novel enzymes with stable and robust properties. These properties unlock an entirely new array of biochemical capabilities for CAR reactions particularly in the applications of high temperature biosynthesis. Additionally, stable AncCARs may prove useful for future enzyme engineering studies with this enzyme. We show that ancestral reconstruction with multiple algorithms offers an important engineering technology for large and/or poorly understood protein families.

## Methods

**Sequence handling.** Unless specified, all algorithms were performed under default settings. Multiple sequence alignments were performed in Geneious version 10.0.2 with MUSCLE[71,72]. The resulting alignments were modified manually. These were then further modified by either: (a) manually removing insertions represented by one, or very few leaves; and (b) the GBlocks algorithm in the Phylogeny.fr program suite[73,74], forming two distinct alignment datasets. Best-fit models of amino acid replacement were identified using ProtTest version 3.4 (ref. [44]). The GBlocks curated alignment was subject to phylogenetic analysis within MrBayes version 3.2.6 (ref. [75]), under the WAG + I + G model of amino acid substitution[45], with two parallel runs of 250,000 Metropolis Coupled Markov-Chain Monte Carlo generations with an independent gamma calculated for all lineages, each with four chains with the heat prior set to 0.02, sampled every 100 generations, with a burn-in of 25%, and all sequences bar those from *Tsukamuraella* and *Segnilliparus* set as the ingroup prior.

ASR was conducted with FastML[39], PAML[38], and Ancescon[43] using the manually curated alignment and the MrBayes tree as inputs. Marginal reconstructions conducted in FastML and PAML were run with the most optimal model available previously defined by ProtTest. PAML was run with eight gamma

rate categories with estimated shape parameters for α, κ, and ω priors. FastML was run with optimization of branch lengths and binary maximum likelihood-based indel reconstruction. Ancescon requires a polytomous root in the input tree: therefore, the MrBayes derived tree had a false polytomy introduced manually in its Newick file. Marginal reconstructions in Ancescon were run with ML-based rate factors and an alignment-based PI vector. Most likely output sequences for each algorithm were aligned in Geneious using MUSCLE. Indels derived from either Ancescon or FastML were transposed to the PAML sequences, producing four final sequences: AncCAR-A, AncCAR-F, AncCAR-PA, and AncCAR-PF.

All sequences are available as supplementary documents.

**Homology modeling of CARs.** The ancestral CARs were modeled using YASARA v.17.8.15 (ref. [76]). The models were based on the structures of the A/T domains of CARs from *Segniliparus rugosus* (PDB ID: 5MST) and *Nocardia iowensis* (PDB ID: 5MSD); and the R domains of CARs from *Mycobacterium marinum* (PDB ID: 5MSO) and *Segniliparus rugosus* (PDB ID: 5MSP)[19]. The alignments used for the ancestral reconstruction were used to direct the homology modeling. Modeling was performed using the default "hmbuild" algorithm. In each case, the preferred model was selected. Images of protein structures were prepared using PyMOL v. 2.0 (Schrödinger)[77]. Root mean squared values for alpha-carbon atom position in the modeled structures were calculated in PyMOL by calculating the best alignment without transform over 10 cycles.

**Purification and storage.** Sequences derived from ASR were modified to contain a 6xHis-tag at the N-terminus. Sequences were codon optimized for *Escherichia coli* K12, and synthesized in two sections. The first sections were synthesized into the pNic28-BSA4 expression vector[78] by Twist Bioscience. The second sections were synthesized by Twist bioscience into their stock vector. The second sections were ligated with the first section and vector by restriction cloning, and sequences verified by Sanger sequencing (Source Bioscience; plasmid sequences available as Supplementary Information). These were co-transformed into BL21(DE3) *E. coli* alongside a pCDF-Duet1 vector containing *Bacillus subtillus* phosphopantetheine transferase[10].

Expression was carried out in LB media supplemented with 150 μM IPTG at 20 °C overnight. Cells were harvested in 20 mM Tris-HCl pH 7.5 with 0.5 M NaCl and 10 mM imidazole and lysed by sonication. The lysate was clarified by centrifugation at 24,000*g*. AncCARs were purified from the soluble fraction by nickel affinity using an ÄKTAXpress (GE Healthcare) using a 1 mL His-Trap FF crude column (GE Healthcare), followed by size exclusion with a Superdex 200 HiLoad 16/60 gel filtration column (GE Healthcare). The nickel affinity column was equilibrated and washed with the cell lysis buffer, and the purified proteins eluted with cell lysis buffer supplemented with 250 mM imidazole. The size exclusion column was eluted with 0.5 M NaCl in 10 mM HEPES-NaOH pH 7.5. The purified proteins were analyzed by SDS-PAGE using 4–12% precast gels run in MOPS buffer (Genscript). Protein concentration was determined using a Nanodrop N2000c nanospectrophotometer (Thermo). If required samples were concentrated to between 0.25 and 0.5 mg mL$^{-1}$ using Vivaspin 6 mL columns with a molecular weight cut-off of 10 kDa (Generon) and stored in 20% (v/v) glycerol at −20 °C. Protein was buffer exchanged into reaction buffer using PD10 desalting columns (Generon) before enzymatic analysis.

**Enzyme assays: standard conditions.** All assays were performed in Grenier flat-bottomed 96-well microtitre plates. Assays were modified from those in ref. [10]. Unless otherwise specified, samples were assayed in triplicate in a 200 μL reaction containing 125 mM HEPES-NaOH (pH 7.5), 1.2 mM ATP, 10 mM MgCl$_2$, 250 μM NADPH, 5 mM substrate, and 5 μg enzyme. Working stocks of each assay component were dissolved in 50 mM HEPES-NaOH (pH 7.5). Their pH was modified to 7.5 to ensure consistent pH across serial dilutions, and volume was made up to 50 mM final concentration of HEPES with MilliQ water. Where necessary, substrates were dissolved in concentrations of DMSO up to 10% (v/v) final reaction in 200 mM HEPES pH 7.5. To begin the reaction, 100 μL substrate working stock in assay buffer was added to 100 μL of a master mix containing the remaining components. Each assay contained substrate buffer solution without substrate in triplicate for blank subtraction of native NADPH degradation rates. Enzyme activity was monitored at 30 °C by measuring the absorbance at 340 nm in a Tecan Infinite 200Pro plate reader in continuous cycles over the course of 10 min with 10 flashes per-well, or using a ThermoFisher SkanIt Pro plate reader in continuous cycles over 10 min. Data were processed in Microsoft Excel and Graphpad Prism v8.2. Experimental data were fit to the Michaelis–Menten equation following calculation of NADPH conversion based on an NADPH standard curve (Supplementary Fig. 16).

**Buffer optimization.** HEPES and Tris were prepared to pH 7.5 at 50, 75, 100, 125, 150, and 275 mM. AncCARs were buffer exchanged into each buffer. AncCAR activity was tested against (*E*)-3-phenylprop-2-enoic acid. All reaction components were prepared in corresponding buffers.

**Analysis of solvent stability.** (*E*)-3-phenylprop-2-enoic acid dissolved in 50 mM HEPES was prepared in 50% (v/v) neat solvent, which was serially diluted in 5 mM

(E)-3-phenylprop-2-enoic acid dissolved in 50 mM HEPES to provide a solvent gradient from 25 to 0% (v/v).

**Analysis of substrate specificity**. CAR activity was tested for each enzyme on 17 aromatic carboxylic acids and 4 aliphatic carboxylic acids. Compounds were prepared to 0.5 M stocks in neat DMSO and diluted to working concentration in assay buffer to a final DMSO concentration of 20%, providing a 10% (v/v) DMSO concentration on the standard assay.

**pH tolerance**. Buffers ranged from pH 3 to 11 in increments of 0.5, prepared at 30 °C. The buffers consisted 50 mM Na-citrate, pH 3.0–5.0; 50 mM MES, pH 5.5–6.5; 50 mM HEPES, pH 7.0–8.0; 50 mM Bicine pH 8.5–9.0; and 50 mM CAPS, pH 9.5–11.0. A series of 80 μL buffer solutions containing 0.25 μg μL$^{-1}$ ancestral protein was constructed and incubated at 30 °C for 30 min. Incubated enzymes were assayed as standard on 5 mM (E)-3-phenylprop-2-enoic acid. Initial rates were calculated as relative activity against acquired rate values at pH 7.5 (100 %). The data were fitted to the following equation[79] to determine the limits of pH tolerance:

$$v = \frac{V_{100}}{\frac{h}{K_1} + 1 + \frac{K_2}{h}},$$

where $V_{100}$ is the maximum rate, $K_1$ and $K_2$ are the proton concentrations where activity drops to 50% at low and high pH respectively, and $h$ is the proton concentration.

**Thermostability following incubation**. In vitro buffer system consisted of standard assay buffer. In vivo-like *Saccharomyces cerevisiae* ion buffer was based on buffer systems described in ref. [56]. Buffer consisted of 50 mM K$_2$HPO$_4$, 75 mM glutamic acid, 85 mM KCl, 10 mM Na$_2$SO$_4$, 2 mM MgCl$_2$, 0.5 mM CaCl$_2$, prepared in 50 mM HEPES and pH modified to 7.5 by adding neat KOH (45%, v/v) dropwise. Salt confirmation buffer was standard assay buffer supplemented with 500 mM NaCl.

Eighty microliters aliquots of each AncCAR at 0.25 μg μL$^{-1}$ in each buffer system were incubated for 30 min at temperatures between 30 and 49 °C, and 50 and 70 °C in a Mastercycler nexus thermocycler (Eppendorf) set to gradient mode. The second aliquot in each gradient was reserved for 80 μL buffer for a negative control. Enzymes were then cooled to 4 °C in the thermocycler for 5 min before being assayed as standard on 5 mM (E)-3-phenylprop-2-enoic acid.

**Differential scanning fluorimetry**. The purest peaks from the size exclusion step of protein purification were buffer exchanged into 50 mM HEPES pH 7.5. Differential scanning fluorimetry (DSF) running mixture was prepared by diluting enzyme to 0.1 μg μL$^{-1}$ to which 10× SYPRO orange was added. DSF was run in sextuplet 20 μL volumes for each condition in a 384-well qPCR plate (Thermo) on a QuantStudio 6 flex real-time PCR machine (Thermo) set to melt-curve mode, with a temperature ramp from 25 to 99 °C ramping at 0.17 °C s$^{-1}$. Data were analyzed using Protein Thermal Shift software v. 1.3.

**Kinetic analysis of CARs on ATP and NADPH**. Enzyme kinetics were assessed by measuring activity of each enzyme on (E)-3-phenylprop-2-enoic acid in the presence of varying concentrations of ATP or NADPH. For both ATP and NADPH titrations, a 1.7× dilution series from 8 mM over 12 points was used. In all instances 800 μM NADPH showed inhibitory effects on CAR activity. Substrate inhibition could also be observed for AncCAR-PA at 470 μM NADPH. Low concentrations of NADPH or ATP caused the reaction to finish quickly meaning concentrations were represented by very few kinetic cycles, resulting in high signal-to-noise ratios. Data for fitting were trimmed of concentrations showing substrate inhibition or high signal-to-noise ratio. Rates were fitted to the Michaelis–Menten equation by non-linear least-squares regression in GraphPad Prism v. 7.

**Kinetic analysis of AncCAR substrate range**. Carboxylic acids were dissolved to near saturation in assay buffer with 20% DMSO. Substrates were titrated in 1.7× dilutions over 8 points. Rates were measured continuously over 6 min in a ThermoFisher MultiSkan GO plate reader in precision mode. Rates were fitted to the Michaelis–Menten model by non-linear least-squares regression in GraphPad Prism v. 8.2.

**Statistics and reproducibility**. Tests for activity of enzymes against substrates were tested using a *t*-test against a no enzyme control. Enzyme samples were tested in triplicate samples, pipetted separately from a common stock. No enzyme controls had at least six replicates pipetted separately from common stocks. Experiments on temperature, pH, and solvent stability were tested in triplicate samples, pipetted separately from a common stock.

**Reporting summary**. Further information on research design is available in the Nature Research Reporting Summary linked to this article.

## Data availability

The research data supporting this publication are openly available from the University of Exeter's institutional repository at: https://doi.org/10.24378/exe.2003[80].

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

## Acknowledgements

We would like to thank the British Biological Research Council's South West Doctoral Training Program for funding and supporting this research. For their support with research methods, we would like to thank Professor Eric Gaucher of Georgia Technological University, and Jennifer Farrar of his research group, for their support and advice with tree building and reconstruction methods. We would finally like to thank Professor Thomas Richards of the University of Exeter for support with the construction of this article, and Alice Cross and Sumita Roy of the Harmer group for their daily support.

## Author contributions

A.T., N.H., and M.v.d.G. conceived the study. A.T. wrote the article. N.H. and M.v.d.G. edited the article. A.T. conducted sequence handling, phylogenetics, A.S.R., protein purification, and assays, and was involved in critical discussion throughout. R.C. conducted protein purification and assays of the proteins, performed protein structure modeling, and contributed to the writing of the article, and was involved in critical discussion throughout. W.F. provided substrates, helped develop the reduction assays, and developed purification protocols for CARs.

## Competing interests

The authors declare no competing interests.
