## [Peer Review File · Communications Biology]

Editorial Note: This manuscript has been previously reviewed at another journal. This document only contains reviewer comments and rebuttal letters for versions considered at *Communications Biology*.

REVIEWERS' COMMENTS:

Reviewer #1 (Remarks to the Author):

The authors have addressed all critical comments, and the revised manuscript can be accepted for publication.

Reviewer #2 (Remarks to the Author):

All my previous concerns are properly addressed in the revised manuscript. I recommend it for publication.

Reviewer #3 (Remarks to the Author):

The authors have addressed most of the concerns raised from my previous review of this manuscript. One minor issue remains. My previous concern about the authors' claims of resurrecting the largest ancient protein missed the mark after my curt statement. Manteca et al. resurrected ancient Titin protein. This is in itself, of course, much larger than CARs. However, the complete Titin was not resurrected, but rather only a fragment. So, while I think AncCAR may be larger than the ancient Titan fragment, the authors need to perform due diligence to confirm their statements. And, at the very least, please cite the Titin study and highlight why your AncCar is (or is not) larger than the ancient Titin fragment. I have no other concerns.

Thomas et al.

Response to reviewers:

We thank the reviewers for their positive comments on our manuscript and constructively critical suggestions at this and past points. We detail our response to their comments below (reviewer comments in blue italics).

REVIEWERS' COMMENTS:

Reviewer #1 (Remarks to the Author):

The authors have addressed all critical comments, and the revised manuscript can be accepted for publication.

We thank the reviewer for their constructive comments on our manuscript.

Reviewer #2 (Remarks to the Author):

All my previous concerns are properly addressed in the revised manuscript. I recommend it for publication.

We thank the reviewer for their constructive comments on our manuscript.

Reviewer #3 (Remarks to the Author):

The authors have addressed most of the concerns raised from my previous review of this manuscript. One minor issue remains. My previous concern about the authors' claims of resurrecting the largest ancient protein missed the mark after my curt statement. Manteca et al. resurrected ancient Titin protein. This is in itself, of course, much larger than CARs. However, the complete Titin was not resurrected, but rather only a fragment. So, while I think AncCAR may be larger than the ancient Titan fragment, the authors need to perform due diligence to confirm their statements. And, at the very least, please cite the Titin study and highlight why your AncCar is (or is not) larger than the ancient Titin fragment. I have no other concerns.

We thank the reviewer for their insight into the larger ancestral reconstruction projects. We have amended the manuscript to state that our reconstruction is *amongst* the largest (rather than the largest), and cited examples of similarly complex reconstructions (including the example suggested by the reviewer). Undertaking further “due diligence” as the referee suggested found one larger recent example so we have modified the text to reflect this.